# Triglyceride and Glucose Index as an Optimal Predictor of Metabolic Syndrome in Lebanese Adults

**DOI:** 10.3390/nu16213718

**Published:** 2024-10-30

**Authors:** Suzan Haidar, Nadine Mahboub, Dimitrios Papandreou, Myriam Abboud, Rana Rizk

**Affiliations:** 1Department of Nutrition and Food Sciences, Lebanese International University, Beirut P.O. Box 146404, Lebanon; suzan.haidar@liu.edu.lb (S.H.); nadine.baltagi@liu.edu.lb (N.M.); 2Department of Clinical Nutrition & Dietetics, College of Health Sciences, University of Sharjah, Sharjah 27272, United Arab Emirates; dpapandreou@sharjah.ac.ae; 3College of Natural and Health Sciences, Zayed University, Dubai 19282, United Arab Emirates; 4Department of Nutrition and Food Science, Lebanese American University, Byblos P.O. Box 36, Lebanon; rana.rizk01@lau.edu.lb; 5Institut National de Santé Publique, d’Epidémiologie Clinique, et de Toxicologie (INSPECT-LB), Beirut P.O. Box 12109, Lebanon

**Keywords:** biomarkers, metabolic syndrome, ROC curve, adults, Lebanon

## Abstract

**Background:** Globally, the prevalence of metabolic syndrome (MetS) is on the rise, especially in Arab countries, which emphasizes the need for reliable ethnic-specific biochemical screening parameters. **Methods:** Two hundred twenty-one Lebanese adults were enrolled in this cross-sectional study. Biochemical parameters including Homeostasis Model Assessment (HOMA), Triglyceride and Glucose index (TyG), ratio of Triglycerides to High-Density Lipoprotein Cholesterol (TG/HDL-C), Atherogenic Index of Plasma (AIP), and Visceral Adiposity Index (VAI) were assessed for their prediction of MetS. Analysis of covariance, logistic regression, expected-versus-observed case ratio were used to determine model calibration, concordance statistic, area under the receiver operating characteristic curve (AUC) and 95% confidence intervals (CIs), sensitivity, specificity, and negative and positive predictive values (PPV, NPV). **Results:** The prevalence of MetS was 44.3%. All biochemical parameters were significantly associated with MetS, with a strong model discrimination (c-statistic between 0.77 and 0.94). In both sex categories, TyG best predicted MetS (females: cut-off value, 8.34; males: cut-off value, 8.43) and showed good estimation among females, but overestimation among males. HOMA had the lowest discriminatory power in both sex categories. **Conclusions:** This study suggests that TyG best predicts MetS, while HOMA has the lowest predictive power. Future larger studies need to focus on harmonizing ethnic specific cut-offs and further validating our results.

## 1. Introduction

Metabolic syndrome (MetS) is a multifaceted collection of metabolic aberrations including elevated waist circumference (WC), insulin resistance, dyslipidemia, and raised blood pressure. MetS increases the risk of development of type II diabetes mellitus (T2DM) and cardiovascular disease (CVD) [1,2]. The prevalence of MetS is escalating globally, particularly in the Arab world. Epidemiological data from the region reveal a substantial burden of this syndrome among its population [3,4,5,6,7]. Notably, Naja et al. [8] reported a prevalence of 42.4% amongst Lebanese aged 18–65 years.

The diagnostic criteria for MetS typically involve a combination of anthropometric, biochemical, and clinical parameters, including the assessment of WC, fasting blood glucose (FBG), triglycerides (TGs), high-density lipoprotein cholesterol (HDL-C), and blood pressure (BP). Insulin resistance (IR) is the most accepted unifying feature contributing to the increased prevalence of MetS [9]. Furthermore, studies have demonstrated that individuals who have IR are more likely to develop T2DM and CVD [10,11]. Raised TG levels, a lower-than-normal HDL-C level, elevated BP, and high BG are considered “identifying abnormalities” of potential carriers of IR [12]. On the other hand, some epidemiologic studies indicate that a substantial proportion of patients with MetS do not have evidence of IR, and the correlation between IR and individual components of MetS is weak to moderate and needs further investigation [13].

Studies examining the knowledge of features of MetS in general and specifically in the East Mediterranean region are scarce. Previous reports indicated difficulties in understanding the perception of the risk factors of MetS especially in rural areas [14] and among college students [15].

There has been increased interest in identifying simple reliable biochemical parameters that can serve as predictors of MetS [16]. The suggested parameters include the Homeostasis Model Assessment (HOMA) that uses the levels of fasting glucose and insulin to assess IR [17], the Triglyceride and Glucose (TyG) index derived from the fasting glucose and TG levels [18], the ratio of TG to HDL-C (TG/HDL-C) and its base 10 logarithm, known as the Atherogenic Index of Plasma (AIP) [19,20], and the Visceral Adiposity Index (VAI), which integrates anthropometric and metabolic parameters [21]. While these biochemical parameters show promising results in predicting MetS, further research is needed to establish their clinical effectiveness, validate their cut-off values, and understand their usability in different ethnic populations. Specifically, one of the challenges in this regard is the lack of ethnic-specific cut-off values. Ethnic-related differences in body composition and metabolism can influence the diagnostic criteria for MetS [22,23]. Failing to account for these differences may lead to underdiagnosis or delayed recognition of MetS. Hence, the establishment of ethnic-specific cut-off values is essential to facilitate a timely diagnosis of MetS. This article aims to explore five emerging biochemical indices, namely, HOMA, TyG, TG/HDL-C, and AIP in addition to VAI as prognosticators of MetS among an adult Lebanese sample.

## 2. Materials and Methods

Lebanese adults, 18–65 years old, were asked to take part in this study through announcements made in local communities and one large university. The inclusion criteria included healthy subjects with no self-reported current infection, not suffering from any disease, and not taking any medications that affect vitamin D metabolism. The study excluded expectant mothers and breastfeeding females.

### 2.1. Ethical Considerations

The study adhered to the guidelines of the “Declaration of Helsinki” and was awarded ethical approval by the Institutional Review Board of the Lebanese International University (LIUIRB-220201-SH-111). Interested individuals were briefed about the study’s objectives and methods and were informed that they had the option to withdraw whenever they desired. Participants provided consent prior to the collection of the data.

### 2.2. Data Collection

Sociodemographic characteristics and medical history information was self-reported by the participants and was collected by research assistants; these data focused on age, gender, educational attainment level, employment status, smoking status, and medical history.

BP: Systolic blood pressure (SBP) and diastolic blood pressure (DBP) were taken two times by experienced technicians by means of a standardized mercury sphygmomanometer. The measurements were taken before blood withdrawal and after ensuring a five-minute rest. The average of the two readings was calculated and used in the analysis.

Anthropometrics: Height (cm) was measured to the nearest 0.1 cm via a stadiometer (ADE, Germany), and mass (kg) to the nearest 100 g while in a standing position with shoes off and wearing light clothes via a beam scale. The body mass index (BMI) was calculated by dividing mass (kg) by squared height in meter (m^2^). Waist circumference (WC, cm) was recorded to the nearest 0.1 cm with a standardized measuring tape at the midpoint between the iliac crest and the lower costal region, on the right side of the body.

Biochemical parameters: After resting for five minutes, a trained licensed phlebotomist drew 5 mL of blood after ensuring that the participants had been fasting for 8 h. Blood was drawn, placed into a sterile tube, and delivered to a licensed lab for analysis via insulated thermal containers. The samples were centrifuged (4000 revolutions/minute) for 10 min. The samples were investigated for fasting insulin levels by an automated chemiluminescence micro-particle immunoassay kit (ARCHITECT; Abbott Laboratories, Abbott Park, IL, USA) and for HDL-C (mg/dL), TG (mg/dL), and FBG (mg/dL) using a Beckman coulter (unicel DXC 600) spectrophotometer.

Biochemical parameters were calculated according to the following:

-HOMA, FBG (mmol/L) × fasting insulin (mU/L) ÷ 22.5 [24];-TyG, Ln [fasting TG (mg/dL) × FBG (mg/dL) ÷ 2] [25];-VAI, males: (WC (cm) ÷ (39.68 +(1.88 ∗ BMI) × (TG ÷ 1.03) ∗ (1.31 ÷ HDL-C); females: (WC (cm) ÷ (36.58 ∗ (BMI × 1.89) ∗ (TG ÷ 0.81) ∗ (1.52 ÷ HDL-C [26];-TG/HDL-C, by dividing the TG level by the HDL-C value;-AIP, log (TG/HDL-C) [27].

### 2.3. Diagnosis of Metabolic Syndrome

MetS was diagnosed as per the International Diabetes Federation (IDF) [1]. Individuals were deemed to have MetS if they had a WC ≥ 94 cm (males) or ≥80 cm (females) or a BMI > 30 kg/m^2^, thus assuming centralized obesity, in addition to two of the following: high TG level (≥150 mg/dL) or taking medication for related treatment; low levels of HDL-C (<40 mg/dL in males and <50 mg/dL in females) or taking medication for related treatment; elevated BP (SBP ≥ 130 or DBP ≥ 85 mmHg) or taking medication to treat hypertension; and FBG ≥100 mg/dL or being diagnosed with non-insulin-dependent diabetes.

### 2.4. Statistical Analysis

The data were analyzed via the Statistical Package for Social Science (SPSS) version 25. A descriptive analysis was performed using means and standard deviations for continuous measurements and frequencies and percentages for categorical variables. The variables were normally distributed, inspected visually via histograms. An Analysis of Covariance (ANCOVA) was conducted to demonstrate the difference in the means of biochemical parameters in relation to the existence and absence of MetS, adjusted for covariates (age, sex, level of education, socioeconomic level, smoking, and family history of diabetes/dyslipidemia/hypertension). Logistic regression models were used with presence/absence of MetS taken as the dependent variable, and anthropometric indices as independent variables, adjusted for sociodemographic and lifestyle characteristics and family history of diseases. Concordance statistic (c-statistic) ranging from 0.5 (chance or no discrimination) to 1.0 (perfect discrimination) was calculated. A calibration investigation was performed to determine the ability of the biochemical parameters to estimate MetS. Expected-versus-observed case ratios (E/O ratios) were calculated to inform the calibration, whereby a value of one indicated a perfect model fit [28]. Finally, the ability of the examined biochemical parameters to predict MetS was determined using the area under the receiver operating characteristic curve (AUC ROC curve) and 95% confidence intervals (CIs), in addition to sensitivity and specificity, and negative and positive predictive value measures. We conducted these analyses for the entire sample and for both males and females. A *p*-value less than 0.05 was regarded as statistically significant.

## 3. Results

In total, 221 participants consented to take part in this study. Almost half of the subjects had MetS (44.3%). The average age was 43.3 ± 16.0 years; most of the subjects were females (62.9%), 55.7% were married and had a low socioeconomic status (50.2%), and 46.6% were employed. Only 21.3% of the subjects smoked cigarettes, while 31.7% smoked the waterpipe. In terms of medical conditions, 18.1% of the participants were diabetics, 28.5% were dyslipidemic, and 20.8% were hypertensive. Additionally, 38.5% had a family history of dyslipidemia (38.5%), hypertension (57.0%), and diabetes (54.3%) (Table 1).

### Description of the Biochemical Parameters

The range, median, mean, and standard deviation of the biochemical parameters stratified according to sex are presented in Table 2.

Table 3 displays the association between the biochemical parameters and MetS adjusted for covariates (age, sex education level, socioeconomic status, smoking, family history of diabetes, family history of dyslipidemia, and family history of hypertension). The findings revealed a significant association between MetS and all the parameters in the total sample and in both sex categories. Significantly higher mean HOMA, TyG, VAI, TG/HDL-C, and AIP were found among those having MetS compared with those who did not have MetS in the total sample and in both sex categories.

The results of logistic regression analysis with MetS being the dependent variable for the total sample and after sex stratification are presented in Table 4. Being a female was significantly associated with MetS. In the total sample and in females, when considering the sociodemographic characteristics as independent variables, the results revealed that a higher age and having a university education were significantly associated with MetS. Amongst males, a higher age was significantly associated with the incidence of MetS. Regarding the biochemical parameters, for the total sample and both sex classes, all measures were significantly associated with MetS.

When considering the c-statistic, the highest difference in c-statistic for males was 0.94 − 0.77 = 0.17, while for females it was 0.91 − 0.84 = 0.073. For the total sample, the highest c-statistic was found for TyG (0.93), followed by AIP (0.92), and the lowest values were found for HOMA (0.88). When adding the anthropometric indices, the discrimination was improved, as shown by the reported c-statistic. For the sociodemographic variables, the c-statistic was 0.83 when adding the anthropometric measures; the c-statistic was improved to the highest level of 0.93.

For females, the highest c-statistic was found for TyG (0.91), followed by VAI (0.90), and the lowest values were found for HOMA (0.89). When the anthropometric indices were added, the discrimination was improved from 0.84 (sociodemographic variables) to 0.91 TyG).

For males, the highest c-statistic was found for TyG (0.948), followed by AIP (0.946), and the lowest values were found for HOMA (0.85). When adding the anthropometrical indices, discrimination was improved from 0.77 (sociodemographic variables) to 0.94 (TyG).

The discrimination power of the sociodemographic variables and biochemical parameters to predict Mets in the total sample and according to gender is presented in Figure 1. The c-statistic for each of the biochemical parameters for predicting Mets in the total sample and in both genders is shown in the Appendix A.

Table 5 displays the calibration analysis of expected and observed cases to estimate the presence of Mets. The ratio between expected and observed cases showed an adequate calibration (E/O ratio between 0.80 and 1.08), and the respective calibration plot revealed no significant over- or underestimation of predictor effects in the total sample (Figure 2) and in both genders (Appendix A).

The diagnostic accuracies of each parameter for the presence of MetS in the total sample and in both sex categories are described in Table 6. For the total sample, the results showed that TyG had the highest AUC values for MetS (AUC: 0.87, 95% CI: 0.82–0.91), where sensitivity was 0.80, and specificity was 0.76 with a cut-off value of 8.41, a PPV of 73.09%, and an NPV of 83.20%. HOMA had the lowest AUC values for MetS (AUC: 0.71, 95% CI: 0.64–0.77), where sensitivity was 0.67, and specificity was 0.60 with a cut-off value of 2.42, a PPV of 57.35%, and an NPV of 69.83%. Among females, the results showed that the TyG index had the highest AUC values for MetS (AUC: 0.84, 95% CI: 0.77–0.90), where sensitivity was 0.81, and specificity was 0.71 with a cut-off value of 8.34, a PPV of 60.64%, and an NPV of 87.63%. HOMA had the lowest AUC values for MetS (AUC: 0.71, 95% CI: 0.62–0.80), where sensitivity was 0.73, and specificity was 0.58 with a cut-off value of 2.46, a PPV of 49.39%, and an NPV of 80.29%. Among males, the results showed that TyG had the highest AUC values for MetS (AUC: 0.91, 95% CI: 0.84–0.97), where sensitivity was 0.83, and specificity was 0.82 with a cut-off value of 8.43, a PPV of 87.37%, and an NPV of 77.18%. HOMA had the lowest AUC values for MetS (AUC: 0.75, 95% CI: 0.64–0.85), where sensitivity was 0.71, and specificity was 0.60 with a cut-off value of 2.12, a PPV of 72.94%, and an NPV of 58.75%. The discrimination power of the biochemical parameters to predict MetS in the total sample and according to sex is displayed in Figure 1.

## 4. Discussion

In this study, we explored whether HOMA, TyG index, VAI, AIP, and TG/HDL-C index predicted the occurrence of MetS. Furthermore, we investigated which measurement had the best predictive power for the syndrome. The study results suggest that TyG best predicted MetS, while HOMA had the lowest predictive power.

Our findings are in line with earlier studies that revealed a high association between the TyG index and MetS, whereby this index outperformed other risk factors and indices for the diagnosis of CVD and IR [29]. Additionally, while evaluating its effectiveness in detecting metabolically obese individuals who have a healthy bodyweight, TyG strongly predicted MetS, with an AUC between 0.855 for males and 0.868 for females [30]. Another study conducted in a Chinese sample also supports the predictive capacity of TyG, with AUC of 0.863 and 0.867 for males and females, respectively [31]. Furthermore, when adolescents of different ethnicities were studied (Korean, Mexican American, and non-Hispanic White adolescents), all ethnicities had significantly higher TyG indices associated with a higher prevalence of MetS when compared to non-Hispanic Black adolescents. The authors reported that those results were due to the fact that the non-Hispanic Black subjects had better lipid profiles [32,33].

Furthermore, the study of different ethnicities and their association with the components of MetS revealed that African Americans were less likely to have raised TG levels when compared to non-Hispanic White subjects. The association between TGs and other components of MetS seemed to be similar for both African Americans and non-Hispanic individuals. Additionally, the authors found a significant association between TG level and WC amongst White females but not amongst African American females when adjusting for demographics and other variables. Among subjects with TG levels below 150 mg/dL, African American females had a higher prevalence of abdominal obesity, raised BP, decreased HDL-C level, and increased BG level and HOMA. Among males, the prevalence rates of elevated BP, raised fasting glucose level, and HOMA were significantly higher among African Americans than among White individuals [34,35].

Moreover, when comparing Whites, Hispanics, and African Americans, the TG/HDL-C ratio was associated with IR particularly among White obese children [34,35], and therefore, it is plausible to use as an indicator to pinpoint those with a higher risk of IR.

Similarly, we suggest the VAI, TG/HDL-C, and AIP as potential screening indices for MetS. The VAI was found to be strongly associated with all criteria of MetS [36]. Furthermore, and coherent with our results, it was shown that the AIP, as an independent parameter, can be used to predict MetS in both males and females with higher accuracy than that achieved using other lipid ratios [37]. In contrast to our results, Zhang et al. [38] demonstrated a higher correlation of AIP with MetS among males but not among females [38]. This could be because WC, smoking rate, and alcohol ingestion were significantly higher for males when compared with females. Additionally, the TG/HDL-C ratio has been recognized as a reliable indicator of IR in overweight patients. According to the Framingham 10-year coronary assessment, elderly patients in health facilities in Naples with TG/HDL ratios ≥ 3.0 were at a more elevated cardiovascular risk compared with those with lower values [39]. Salazar et al. [40] showed that TG/HDL-C can be used to identify people with insulin resistance just as accurately as for MetS diagnosis [40]. However, the use of the TG/HDL-C ratio to identify those with insulin resistance with increased cardiometabolic risk is complex, given that effectiveness varies among different ethnicities [41]. Non-Hispanic Whites and Mexican Americans had a higher optimal cutoff for TG/HDL-C to determine hyperinsulinemia compared with non-Hispanic Blacks [42]. Moreover, TG/HDL-C, in Black adults, was not significantly associated with insulin resistance. This finding was attributed to racial variations in HDL-C, TG, and fasting serum insulin concentrations [43,44].

Finally, a surprising finding in our study was that HOMA had the least predictive ability for MetS, which is a result that is not in line with research performed by Hanley and colleagues, who found that the risk of MetS rises with increased HOMA values [45]. Additionally, Esteghamati et al. reported that IR and MetS were significantly associated with HOMA [46]. However, in their study, the TyG index outperformed HOMA in identifying MetS. This was also found in other studies [29,47]. Although the exact mechanism for the superiority of the TyG index above HOMA in predicting MetS is not well understood, some hypotheses have been formulated [48,49]. First, HOMA mostly reflects hepatic insulin resistance, while the TyG index reflects both hepatic and muscular insulin resistance [50]. Raised serum TG levels interfere with how glucose is metabolized in muscles, resulting in insulin resistance [51]. Additionally, visceral fat-derived hypertriglyceridemia increases the free fatty acid levels hepatically, which reduces insulin sensitivity and raises the hepatic glucose output [52]. Conversely, both insulin and glucose values are needed to compute HOMA. Insulin resistance is mostly found in the liver because of the basal insulin-dependent suppression of the hepatic glucose output in a fasting state [53]. It is also plausible that elevated TG levels reflect the systematic effect of inflammation on insulin resistance [54]. Tumor necrosis factor (TNF) raises the serum TG levels by triggering the synthesis of free fatty acids and inhibiting the hepatic absorption of TG-rich lipoproteins [55]. Therefore, compared with HOMA, the TyG index may be more closely associated with chronic inflammation [56]. Finally, a likely reason for the superiority of TyG is that it takes into account two components of MetS necessary for its diagnosis (triglycerides and glucose), whereas the HOMA formula only assesses one component (glucose) [29]. This is also a valid point when considering the VAI, as its calculation is based on three components of MetS (TG, HDL-C, and WC). Additionally, TyG is a less costly parameter than HOMA; so, the use of HOMA might be limited by its cost, as it necessitates insulin standardization [57].

As far as we are aware, this particular study is the first of its kind conducted in the Middle East evaluating novel indices for the prediction of MetS. Another strength is that all the assessment tools used in this research were validated, and the data collectors were well-trained dietitians, licensed phlebotomists, and nurses. Finally, blood analysis was conducted at an accredited medical laboratory, where, for example for lipids and FBG, the instruments, operated and maintained according to the manufacturers’ instructions, should exhibit a within-run coefficient of variation of ≤2.0% for all sample levels. However, limitations of this study are the possibility of self-selection bias, since the participants were volunteers who answered our community announcement, and the self-reported data for the inclusion and exclusion criteria and for some of the participants’ characteristics. Furthermore, there was a preponderance of female volunteers. This is often the case in studies based on volunteer recruitment. It is worthy to note that this is not a nationally representative study and that the current research was performed on a relatively small sample that was recruited from one large university and nearby local communities. Hence, our results, specifically, the reported prevalence of MetS and the cut-off points identified, may not be generalizable. Future studies should focus on recruiting a larger, more representative sample and should account for other possible predictors such as food intake, alcohol consumption, and physical activity.

## 5. Conclusions

Our findings hold clinical significance for preventive public health strategies for the early and uncostly detection of MetS, even if the patient is not physically present, so that anthropometric measurements cannot be taken. TyG could be used as an early detector for MetS during a routine biochemical screening. Future larger studies need to focus on harmonizing ethnic-specific cut-offs and further validating our results.

## Figures and Tables

**Figure 1 nutrients-16-03718-f001:**
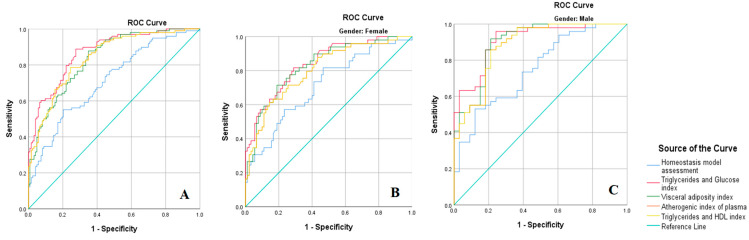
Discriminatory power of biochemical parameters to predict metabolic syndrome. (**A**) ROC curves illustrating the discriminatory power of the examined biochemical parameters to predict metabolic syndrome in the total sample. (**B**) ROC curves illustrating the discriminatory power of the examined biochemical parameters to predict metabolic syndrome among females. (**C**) ROC curves illustrating discriminatory power of the examined biochemical parameters to predict metabolic syndrome among males.

**Figure 2 nutrients-16-03718-f002:**
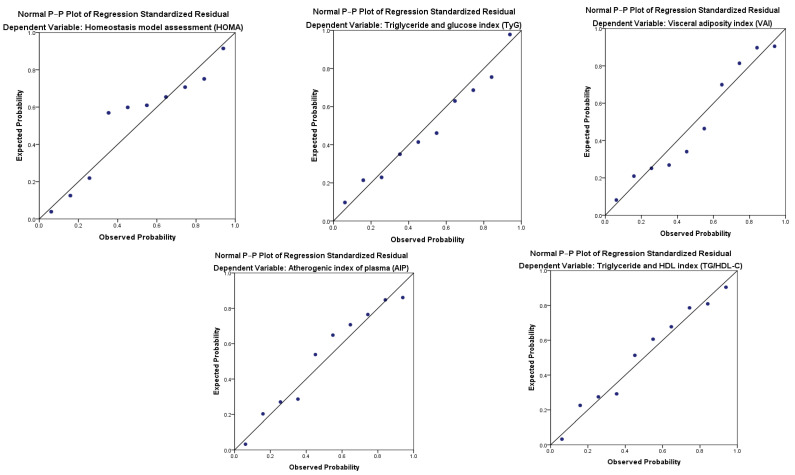
Calibration plots of the biochemical parameters to predict metabolic syndrome based on the contingency table for the Hosmer–Lemeshow statistic for the total sample.

**Table 1 nutrients-16-03718-t001:** Sociodemographic and other characteristics of the participants (N = 221).

Characteristics	Total Sample	Females (n = 139; 62.9%)	Males (n = 82; 37.1%)
**Mean age in years**	43.3 (SD, 16.0)	42.6 (SD, 16.0)	44.5 (SD, 16.1)
**Body Mass Index in kg/m^2^**	28.43 (SD, 6.10)	28.86 (SD, 6.57)	27.69 (SD, 5.15)
**Fasting Blood Glucose in mg/dL**	97.78 (SD, 30.75)	94.58 (SD, 22.15)	103.20 (SD, 41.04)
**Systolic Blood Pressure in mmHg**	116.41 (SD, 15.90)	114.20 (SD, 16.67)	120.17 (SD, 13.80)
**Diastolic Blood Pressure in mmHg**	75.62 (11.21)	74.22 (SD, 10.84)	78.00 (SD, 11.51)
**Waist Circumference in cm**	96.31 (SD, 16.58)	93.92 (SD, 17.60)	100.37 (SD, 13.88)
**Triglycerides in mg/dL**	114.00 (SD, 78.58)	106.63 (SD, 67.36)	126.50 (SD, 93.76)
**High-Density Lipoprotein Cholesterol in mg/dL**	46.90 (SD, 13.02)	50.58 (SD, 13.23)	40.65 (SD, 9.99)
**Marital status** (Married)	123 (55.7%)	65 (46.8%)	58 (70.7%)
**Education level**			
University degree	102 (46.2%)	59 (42.4%)	43 (52.4%)
High school	41 (18.6%)	29 (20.9%)	12 (14.6%)
Middle education	37 (16.7%)	25 (18.0%)	12 (14.6%)
Primary education	30 (13.6%)	15 (10.8%)	15 (18.3%)
Illiterate	11 (5.0%)	11 (7.9%)	-
**Socioeconomic status**			
Low	111 (50.2%)	72 (52.2%)	39 (47.6%)
Medium	102 (46.2%)	63 (45.7%)	39 (47.6%)
High	7 (3.2%)	3 (2.2%)	4 (4.9%)
**Employment status** (employed)	103 (46.6%)	46 (33.6%)	57 (69.5%)
**Cigarette smoker**	47 (21.3%)	27 (19.4%)	20 (24.4%)
**Waterpipe smoker**	70 (31.7%)	49 (35.3%)	21 (25.6%)
**Presence of Metabolic Syndrome**	98 (44.3%)	49 (35.3%)	49 (59.8%)
**Comorbidities**			
Personal history of diabetes	40 (18.1%)	26 (18.7%)	14 (17.1%)
Family history of diabetes	120 (54.3%)	76 (55.1%)	44 (53.7%)
Personal history of dyslipidemia	63 (28.5%)	43 (30.9%)	20 (24.4%)
Family history of dyslipidemia	85 (38.5%)	58 (42.0%)	27 (32.9%)
Personal history of hypertension	46 (20.8%)	27 (19.4%)	19 (23.2%)
Family history of hypertension	126 (57.0%)	83 (59.7%)	43 (53.1%)

**Table 2 nutrients-16-03718-t002:** Description of the biochemical parameters.

Females (N = 139)				
	Mean ± SD	Median	Min	Max
Homeostasis model assessment (HOMA)	3.03 ± 2.06	2.56	0.44	12.57
Triglyceride and glucose index (TyG)	8.34 ± 0.65	8.29	6.88	10.60
Visceral adiposity index (VAI)	2.06 ± 1.75	1.49	0.33	11.21
Atherogenic index of plasma (AIP)	0.27 ± 0.30	0.25	−0.37	1.08
Triglyceride and high-density lipoprotein cholesterol index (TG/HDL-C)	2.39 ± 1.89	1.78	0.42	12.05
**Males (N = 82)**				
	**Mean ± SD**	**Median**	**Min**	**Max**
Homeostasis model assessment (HOMA)	3.06 ± 2.56	2.43	0.49	16.90
Triglyceride and glucose index (TyG)	8.53 ± 0.73	8.55	6.99	10.37
Visceral adiposity index (VAI)	2.19 ± 2.27	1.45	0.31	16.08
Atherogenic index of plasma (AIP)	0.41 ± 0.33	0.39	−0.24	1.41
Triglyceride and high-density lipoprotein cholesterol index (TG/HDL-C)	3.58 ± 3.64	2.48	0.57	25.60

**Table 3 nutrients-16-03718-t003:** Association between biochemical indices and metabolic syndrome.

	Total Sample			Females			Males		
	Presence of MetS (n = 98)	Absence of MetS (n = 123)		Presence of MetS (n = 49)	Absence of MetS (n = 90)		Presence of MetS (n = 49)	Absence of MetS (n = 33)	
	Mean ± SE	Mean ± SE	MD (95% CI)	Mean ± SE	Mean ± SE	MD (95% CI)	Mean ± SE	Mean ± SE	MD (95% CI)
**Homeostasis model assessment (HOMA)**	4.15 ± 0.23	2.11 ± 0.19	2.03 (1.38; 2.69)	4.41 ± 0.31	2.30 ± 0.21	2.11 (1.29; 2.93)	3.67 ± 0.35	1.90 ± 0.43	1.77 (0.58; 2.95)
*p*-value	**<0.001**			**<0.001**			**<0.001**		
**Triglyceride and glucose index (TyG)**	8.83 ± 0.06	8.08 ± 0.05	0.74 (0.56; 0.92)	8.77 ± 0.09	8.11 ± 0.06	0.66 (0.42; 0.90)	8.89 ± 0.08	8.01 ± 0.10	0.88 (0.59; 1.17)
*p*-value	**<0.001**			**<0.001**			**<0.001**		
**Visceral adiposity index (VAI)**	3.22 ± 0.20	1.28 ± 0.17	1.93 (1.37; 2.50)	3.31 ± 0.25	1.43 ± 0.17	1.87 (1.21; 2.54)	2.99 ± 0.32	1.10 ± 0.39	1.89 (0.80; 2.97)
*p*-value	**<0.001**			**<0.001**			**<0.001**		
**Atherogenic index of plasma (AIP)**	0.51 ± 0.03	0.17 ± 0.02	0.33 (0.25; 0.42)	0.45 ± 0.04	0.17 ± 0.03	0.28 (0.16; 0.39)	0.58 ± 0.04	0.17 ± 0.05	0.41 (0.28; 0.55)
*p*-value	**<0.001**			**<0.001**			**<0.001**		
**Triglyceride and HDL index (TG/HDL-C)**	4.24 ± 0.28	1.77 ± 0.23	2.46 (1.68; 3.25)	3.67 ± 0.28	1.73 ± 0.18	1.94 (1.21; 2.66)	4.84 ± 0.51	1.82 ± 0.63	3.02 (1.28; 4.76)
*p*-value	**<0.001**			**<0.001**			**<0.001**		

MetS: metabolic syndrome; SE: standard error; MD: mean difference; CI: confidence interval; the results were adjusted for age, sex, education level, socioeconomic status, smoking, family history of diabetes, family history of dyslipidemia, and family history of hypertension. A Bonferroni correction was performed to correct the *p*-value for the multiple predictors used, whereby a significant *p*-value was α/8, i.e., <0.0045.

**Table 4 nutrients-16-03718-t004:** Logistic regression analysis considering the presence of metabolic syndrome as the dependent variable.

		Dependent Variable: Presence vs. Absence of Mets								
		Total Sample			Females			Males		
		ORa (95% CI)	c Statistic	*p*-Value	ORa (95% CI)	c Statistic	*p*-Value	ORa (95% CI)	c Statistic	*p*-Value
	**Independent Variables**									
	**Sociodemographic Variables**									
Model 1	Age	1.08 (1.05; 1.10)	0.830	**<0.001**	1.08 (1.04; 1.12)	0.841	**<0.001**	1.09 (1.04; 1.15)	0.773	**<0.001**
	Sex (female vs. male *)	3.85 (1.85; 7.99)		**<0.001**	-			-		-
	Education level (university level vs. illiterate *)	0.16 (0.03; 0.78)		**0.024**	0.11 (0.02; 0.65)		**0.015**	0.56 (0.10; 2.88)		0.490
	Marital status (married vs. single *)	0.84 (0.40; 1.77)		0.663	1.44 (0.57; 3.62)		0.431	0.18 (0.03; 1.07)		0.060
	Smoking status (yes vs. no *)	0.90 (0.46; 1.76)		0.771	1.03 (0.40; 2.62)		0.945	0.90 (0.30; 2.67)		0.857
	Family history of diabetes (yes vs. no *)	1.32 (0.66; 2.63)		0.423	1.42 (0.56; 3.55)		0.454	1.89 (0.56; 6.36)		0.303
	Family history of dyslipidemia (yes vs. no *)	1.48 (0.88; 2.48)		0.134	1.78 (0.85; 3.72)		0.122	0.85 (0.36; 2.02)		0.725
	Family history of hypertension (yes vs. no *)	0.87 (0.46; 1.61)		0.658	1.11 (0.46; 2.71)		0.805	0.81 (0.31; 2.14)		0.682
	**Biochemical parameters ***									
Model 2	Homeostasis model assessment (HOMA)	4.21 (2.31; 7.67)	0.887	**<0.001**	2.59 (1.61; 4.18)	0.893	**<0.001**	7.07 (2.23; 22.45)	0.850	**0.001**
Model 3	Triglyceride and glucose index (TyG)	10.94 (5.13; 23.30)	0.930	**<0.001**	6.89 (3.47; 13.67)	0.913	**<0.001**	17.04 (4.93; 58.80)	0.948	**<0.001**
Model 4	Visceral adiposity index (VAI)	12.14 (5.09; 28.97)	0.919	**<0.001**	5.80 (2.97; 11.35)	0.902	**<0.001**	149.56 (13.16; 1698.81)	0.939	**<0.001**
Model 5	Atherogenic index of plasma (AIP)	8.16 (4.20; 15.87)	0.923	**<0.001**	4.38 (2.54; 7.55)	0.898	**<0.001**	10.05 (3.60; 28.05)	0.946	**<0.001**
Model 6	Triglyceride and HDL-C index (TG/HDL-C)	9.45 (4.29; 20.80)	0.921	**<0.001**	4.64 (2.54; 8.47)	0.899	**<0.001**	81.89 (9.34; 718.01)	0.938	**<0.001**

ORa: adjusted odds ratio. * The models for the total sample were adjusted for the sociodemographic characteristics age, sex, education level, marital status, smoking status, family history of diabetes, family history of dyslipidemia, and family history of hypertension.

**Table 5 nutrients-16-03718-t005:** Calibration in the presence of Mets for the overall sample and after gender stratification.

	Expected Cases ^a^ N	Observed Cases N	E/O Ratio
**Overall**			
Homeostasis model assessment (HOMA)	87	94	0.92
Triglyceride and glucose index (TyG)	92	94	0.97
Visceral adiposity index (VAI)	90	94	0.95
Atherogenic index of plasma (AIP)	94	94	1.00
Triglyceride and HDL-C index (TG/HDL-C)	93	94	0.98
**Female**			
Homeostasis model assessment (HOMA)	38	47	0.80
Triglyceride and glucose index (TyG)	43	47	0.91
Visceral adiposity index (VAI)	41	47	0.87
Atherogenic index of plasma (AIP)	42	47	0.89
Triglyceride and HDL-C index (TG/HDL-C)	44	47	0.93
**Male**			
Homeostasis model assessment (HOMA)	50	47	1.06
Triglyceride and glucose index (TyG)	48	47	1.02
Visceral adiposity index (VAI)	48	47	1.02
Atherogenic index of plasma (AIP)	51	47	1.08
Triglyceride and HDL-C index (TG/HDL-C)	49	47	1.04

E/O, ratio of expected to observed cases. ^a^ Expected and observed cases were derived from the classification table in the logistic regression model.

**Table 6 nutrients-16-03718-t006:** Measures of the diagnostic accuracy of each parameter in relation to the presence of metabolic syndrome.

	Value	Sensitivity	Specificity	PPV	NPV
**Total Sample**					
Homeostasis model assessment (HOMA)	2.42	67.3	60.2	57.35	69.83
Triglyceride and glucose index (TyG)	8.41	80.6	76.4	73.09	83.20
Visceral adiposity index (VAI)	1.49	76.5	70.7	67.50	79.09
Atherogenic index of plasma (AIP)	0.29	78.6	70.7	68.09	80.60
Triglyceride and HDL-C index (TG/HDL-C)	1.95	78.6	70.7	68.09	80.60
**Females**					
Homeostasis model assessment (HOMA)	2.46	73.5	58.9	49.39	80.29
Triglyceride and glucose index (TyG)	8.34	81.6	71.1	60.64	87.63
Visceral adiposity index (VAI)	1.61	77.6	71.1	59.43	85.33
Atherogenic index of plasma (AIP)	0.29	71.4	71.1	57.41	82.00
Triglyceride and HDL-C index (TG/HDL-C)	1.97	71.4	71.1	57.41	82.00
**Males**					
Homeostasis model assessment (HOMA)	2.12	71.4	60.6	72.94	58.75
Triglyceride and glucose index (TyG)	8.43	83.7	82.0	87.37	77.18
Visceral adiposity index (VAI)	1.27	85.7	81.8	87.51	79.36
Atherogenic index of plasma (AIP)	0.31	85.7	78.8	85.74	78.74
Triglyceride and HDL-C index (TG/HDL-C)	1.79	89.8	72.7	73.03	82.73

## Data Availability

The data presented in this study are available on request from the corresponding author. The data are not publicly available due to confidentiality and privacy concerns.

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
