# Peer review of "Triglyceride and Glucose Index as an Optimal Predictor of Metabolic Syndrome in Lebanese Adults"

_nutrients, 2024, doi:10.3390/nu16213718_

Round 1

Reviewer 1 Report

Comments and Suggestions for Authors

The study examines multiple bochemical markers and their associations in metabolic syndrome. The strength of the paper is statysticsL logistic regressions and c-statistics.

However, the investigated group is focused on Meadle-East population, so especially in the discussion it is worth to compare different ethnic or race gropus. What is more relatively small sample size limit the power of statistics.  Additionally limits are that participants were volontiers so it may brings self-selection bias.

Please update references. Only 5 articles are new. You may adress in dissucion  other specyfic ethnic groups eg. PMID: 31190766 or  PMID: 33913390.

Author Response

The study examines multiple bochemical markers and their associations in metabolic syndrome. The strength of the paper is statysticsL logistic regressions and c-statistics.

However, the investigated group is focused on Meadle-East population, so especially in the discussion it is worth to compare different ethnic or race gropus. What is more relatively small sample size limit the power of statistics.  Additionally limits are that participants were volontiers so it may brings self-selection bias.

Please update references. Only 5 articles are new. You may adress in dissucion  other specyfic ethnic groups eg. PMID: 31190766 or  PMID: 33913390.

Reply: Additional Information on the biochemical markers of metabolic syndrome in different ethnic groups was added in the discussion page 4 lines 256-266 and the text now reads: “Furthermore, studying different ethnicities and their association with the components of the MetS, African Americans had lower prevalence of elevated TG as compared with non-Hispanic whites. The association of TG with other components of the metabolic syndrome appeared to be similar between African Americans and non-Hispanic whites except for one. There was significant association of TG with WC among white women but not among African American women after adjusting for demographic and other variables. In participants with TG < 150 mg/dL, African American women had higher prevalence rates than white women of abdominal obesity, elevated BP, low HDL-C, elevated fasting glucose and homeostasis model assessment of insulin resistance (HOMA-IR). In men, the prevalence rates of high BP, elevated fasting glucose, and HOMA-IR were significantly higher in African Americans than in whites. Moreover, when comparing white, Hispanics and African Americans, The TG/HDL-C ratio was  associated with IR mainly in the white obese boys and girls and thus may be used with other risk factors to identify subjects at increased risk of IR-driven morbidity.”

Additional References:

  • Lin, S. X., Carnethon, M., Szklo, M., & Bertoni, A. (2011). Racial/ethnic differences in the association of triglycerides with other metabolic syndrome components: the Multi-Ethnic Study of Atherosclerosis. Metabolic syndrome and related disorders9(1), 35-40.
  • Giannini, C., Santoro, N., Caprio, S., Kim, G., Lartaud, D., Shaw, M., ... & Weiss, R. (2011). The triglyceride-to-HDL cholesterol ratio: association with insulin resistance in obese youths of different ethnic backgrounds. Diabetes care34(8), 1869-1874.

Reviewer 2 Report

Comments and Suggestions for Authors

The findings appeared to be interesting, while several notations were required appropriately for readers.

1.       The comparative studies previously reported for the insulin resistance-related parameters should be summarized in Introduction. After that, the novel insight of the study could be given to the readers.

2.       Concerning the ethnicity, the knowledge of feature of MetS in Lebanese adults should be described in comparison to that of other countries in Introduction.

3.       Methods; How was the current infection checked in the inclusion to the study?

4.       How many people were excluded from the study?

5.       Were the endocrine diseases checked or excluded? It is important because such diseases produce secondary dyslipidemia and glucose dysregulation.

6.       Triglycerides are easily affected by alcohol and exercise habits. These factors could be considered for the analysis.

7.       Assay performance (variation of coefficient) of lipids and glucose could be disclosure.

8.       Results; Significant digits of the data could be considered (e.g., 2 decimal point places might not be informative for age.

9.       Simply, each lipid and glucose level, in addition to blood pressure, BMI and WC, levels could be disclosure.

10.   Table 3; The data could be shown in not SE but SD if the mean is used.

11.   Discussion; The results might be affected and modified because the study was done in a University.

12.   Row 275; The authors stated about liver states. The data of liver function tests could be shown in the study population.

13.   Once the term (e.g., index) was abbreviated, the repeated abbreviations are not necessary. Please recheck the repetition throughout the text.

14.   Row 18; The word were might be inserted between ‘adults’ and ‘enrolled’.

15.   Row 37; The sentence could have the adequate references.

16.   HDL and HDL-C were mixed in the text. HDL and HDL-C are generally different.

Author Response

The findings appeared to be interesting, while several notations were required appropriately for readers.

  1. The comparative studies previously reported for the insulin resistance-related parameters should be summarized in Introduction. After that, the novel insight of the study could be given to the readers.

Reply: We thank the reviewer for the comment. Additional information has been added to the introduction pages 1-2 lines: 45-51 and now reads as follows: “Insulin resistance (IR) is the most accepted unifying theory contributing to the increased prevalence of MetS. Furthermore, Studies have verified that individuals with IR are more prone to later developing T2DM and CVD. The presence of elevated TG, low HDL-C, high blood pressure and fasting glycemia or high postprandial glycemia as "identifying abnormalities" of possible carriers of IR. On the other hand, some epidemiologic studies indicate that a substantial proportion of patients with the MetS do not have evidence of IR, and the correlation between insulin resistance and individual components of the syndrome is weak to moderate.”

Additional References:

  • Oliveira CL, Mello MT, Cintra IP, Fisberg M. Obesidade e síndrome metabólica na infância e adolescência. Rev Nutr. 2004;17(2):237-45
  • Sung RY, Tong PC, Yu CW, Lau PW, Mok GT, Yam MC, et al. High prevalence of insulin resistance and metabolic syndrome in overweight / obese preadolescent Hong Kong Chinese children aged 9-12 years. Diabetes Care. 2003;26(1):250-1.
  • Weiss R, Dziura J, Burgert TS, Tamborlane W, Taksali SE, Yeckcel CW. Obesity and the metabolic syndrome in children and adolescents. N Engl J Med. 2004;350(23):2362-74.
  • Einhorn D, Reaven GM, Cobin RH, Ford E, Ganda OP, Handelsman Y, et al. American College of Endocrinology position statement on the insulin resistance syndrome. Endocr Pract. 2003;9(3):237-52.
  • Mikhail, N. (2009). The metabolic syndrome: insulin resistance. Current hypertension reports11(2), 156-158.

  1. Concerning the ethnicity, the knowledge of feature of MetS in Lebanese adults should be described in comparison to that of other countries in Introduction.

Reply: Additional information added to the introduction as requested on page 2 lines 52-52 and now the text reads as follows: “Studies examining the knowledge of features of MetS in general and specifically in the East Mediterranean region are scarce. Previous reports indicated difficulties in understanding the perception of the risk factors of MetS especially in rural areas and also among college students.”

Additional References:

  • Joshi, A., Mehta, S., Grover, A., Talati, K., Malhotra, B., & Puricelli Perin, D. M. (2013). Knowledge, attitude, and practices of individuals to prevent and manage metabolic syndrome in an Indian setting. Diabetes Technology & Therapeutics15(8), 644-653.
  • Green JS, Grant M, Hill KL, Brizzolara J, Belmont B: Heart disease risk perception in college men and women. J Am Coll Health. 2003, 51 (5): 207-211. 10.1080/07448480309596352.

  1. Methods; How was the current infection checked in the inclusion to the study?

Reply: It was self-reported. This information was added to the methods section, and recognized in the limitations section.

  1. How many people were excluded from the study?

Reply: All consented participants were included in the study and provided complete data for analysis. Hence, we dd not exclude any participants.

  1. Were the endocrine diseases checked or excluded? It is important because such diseases produce secondary dyslipidemia and glucose dysregulation.

Reply: It was self-reported. This information was added to the methods section, and recognized in the limitations section.

  1. Triglycerides are easily affected by alcohol and exercise habits. These factors could be considered for the analysis.

Reply: Indeed, these are factors that could affect biochemical parameters; the lack of this information is already included in the limitations section.

  1. Assay performance (variation of coefficient) of lipids and glucose could be disclosure.

            Reply: Instruments operated and maintained according to manufacturer’s instructions should exhibit a within-run coefficient of variation of ≤2.0% for all sample levels. We acknowledged that blood analysis was conducted at an accredited medical laboratory in the strengths section and we added this information.

  1. Results; Significant digits of the data could be considered (e.g., 2 decimal point places might not be informative for age.

            Reply: We adjusted for age in the tables and the text. For other biochemical parameters and statistical values, we opted to retain two decimal places, while for p-values, we used three decimal places, as this is the standard reporting practice.

  1. Simply, each lipid and glucose level, in addition to blood pressure, BMI and WC, levels could be disclosure.

            Reply: This information was added to Table 1.

  1. Table 3; The data could be shown in not SE but SD if the mean is used.

Reply: The results were adjusted over several covariates: age, sex, education level, socioeconomic status, smoking, family history of diabetes, family history of dyslipidemia, and family history of hypertension. Therefore, we used Standard Error (SE) and not Standard Deviation (SD).

  1. Discussion; The results might be affected and modified because the study was done in a University.

Reply: We thank the reviewer for this comment. Indeed, this was limitation already acknowledged in our discussion; we made it now more explicit.

  1. Row 275; The authors stated about liver states. The data of liver function tests could be shown in the study population.

Reply: Such information is not reported in Line 275, and we do not address liver function tests throughout the manuscript, as we did not collect such information.

  1. Once the term (e.g., index) was abbreviated, the repeated abbreviations are not necessary. Please recheck the repetition throughout the text.

            Reply: Thank you for the comment. Abbreviations corrected all through the text.

  1. Row 18; The word ‘were’ might be inserted between ‘adults’ and ‘enrolled’.

            Reply: The word was inserted between “adults “and “enrolled”.

  1. Row 37; The sentence could have the adequate references.

            Reply: Additional references Added

            Additional References:

  • Haffner, S. M. (2006). The metabolic syndrome: inflammation, diabetes mellitus, and cardiovascular disease. The American journal of cardiology97(2), 3-11.
  • Aschner, P. (2010). Metabolic syndrome as a risk factor for diabetes. Expert Review of Cardiovascular Therapy8(3), 407-412.
  • Silveira Rossi, J. L., Barbalho, S. M., Reverete de Araujo, R., Bechara, M. D., Sloan, K. P., & Sloan, L. A. (2022). Metabolic syndrome and cardiovascular diseases: Going beyond traditional risk factors. Diabetes/metabolism research and reviews38(3), e3502.
  • Han, T. S., & Lean, M. E. (2016). A clinical perspective of obesity, metabolic syndrome and cardiovascular disease. JRSM cardiovascular disease5, 2048004016633371.

  1. HDL and HDL-C were mixed in the text. HDL and HDL-C are generally different.

             Reply: Thank you for the comment. These terms were fixed in the text.

Round 2

Reviewer 1 Report

Comments and Suggestions for Authors

Please update the refferences

Author Response

We have now updated the ref and marked in red inside the doc.

Reviewer 2 Report

Comments and Suggestions for Authors

The diseases to be excluded might be included because the study was on self-report on diseases. This could be more clearly expressed in the study limitation.

Author Response

We have now added your recommendation to the limitation section